# Validation of the Activ8 Activity Monitor for Monitoring Postures, Motions, Transfers, and Steps of Hospitalized Patients

**DOI:** 10.3390/s24010180

**Published:** 2023-12-28

**Authors:** Marlissa L. Becker, Henri L. P. Hurkmans, Jan A. N. Verhaar, Johannes B. J. Bussmann

**Affiliations:** 1Physical Therapy, Department of Orthopaedics and Sports Medicine, Erasmus MC, University Medical Center Rotterdam, 3000 CA Rotterdam, The Netherlands; 2Department of Orthopaedics and Sports Medicine, Erasmus MC, University Medical Center Rotterdam, 3000 CA Rotterdam, The Netherlands; 3Department of Rehabilitation Medicine, Erasmus MC, University Medical Center Rotterdam, 3000 CA Rotterdam, The Netherlands

**Keywords:** validation, activity monitor, hospitalized, patients, physical behavior

## Abstract

Sedentary behaviors and low physical activity among hospitalized patients have detrimental effects on health and recovery. Wearable activity monitors are a promising tool to promote mobilization and physical activity. However, existing devices have limitations in terms of their outcomes and validity. The Activ8 device was optimized for the hospital setting. This study assessed the concurrent validity of the modified Activ8. Hospital patients performed an activity protocol that included basic (e.g., walking) and functional activities (e.g., room activities), with video recordings serving as the criterion method. The assessed outcomes were time spent walking, standing, upright, sedentary, and newly added elements of steps and transfers. Absolute and relative time differences were calculated, and Wilcoxon and Bland–Altman analyses were conducted. Overall, the observed relative time differences were lower than 2.9% for the basic protocol and 9.6% for the functional protocol. Statistically significant differences were detected in specific categories, including basic standing (*p* < 0.05), upright time (*p* < 0.01), and sedentary time (*p* < 0.01), but they did not exceed the predetermined 10% acceptable threshold. The modified Activ8 device is a valid tool for assessing body postures, motions, steps, and transfer counts in hospitalized patients. This study highlights the potential of wearable activity monitors to accurately monitor and promote PA among hospital patients.

## 1. Introduction

Patient activity levels in hospitals are typically very low, which can have detrimental effects on patients’ health and recovery during their hospital admission [1]. Even when hospital patients are capable of engaging in independent ambulation, they still spend most of their time in sedentary positions, such as sitting or lying [2]. These prolonged periods of inactivity and sedentary behaviors may lead to complications such as a decline in physical functioning and ability to perform ADL activities [3,4], reductions in muscle strength [5], and even cardiovascular fitness deconditioning [6]. Fortunately, these adverse effects can be reduced through targeted interventions such as mobilization and increasing physical activity [7].

Despite the recognized benefits of effective mobilization, several barriers stand in the way of widespread implementation [8]. Firstly, hospital staff are often facing challenging time constraints, which restrict their capacity to facilitate and support effective patient mobilization [9,10]. Secondly, patient compliance is influenced by various factors, such as fatigue, pain, or weakness, which can diminish a patient’s motivation for mobilization or ability to engage in physical activity [11,12]. Lastly, in the hospital setting, existing environmental limitations, such as the presence of restraining IV poles or the lack of suitable equipment, can further impede mobilization, for example, by restricting freedom of movement [11,13]. Nevertheless, a variety of strategies can be implemented to promote mobilization and increase physical activity. These may include educating both healthcare professionals and patients about the importance of mobilization or integrating multidisciplinary teams to enhance patient motivation [9]. Another option is the utilization of technology-assisted interventions, such as wearable activity monitors. These devices offer the possibility of continuously monitoring physical behavior and also provide real-time feedback to support both healthcare professionals and patients.

In the general population, activity monitors are used as a valuable tool for both feedback and motivation [14,15,16,17,18]. However, it is important to note that these devices may not be suitable for monitoring physical behavior in hospital populations for several reasons. Firstly, the goals of monitoring physical behavior in hospital patients differ from those of the general population. While the general population may be primarily interested in metrics like counts, energy expenditure, or exercise intensity, the needs of hospital patients and their healthcare professionals are focused on other components of physical behavior. Specific information, such as time spent in particular postures or movements, the frequency of transfers, and the number of daily steps, is more relevant [8,19]. Secondly, the distinct movement patterns of hospital patients present a hurdle for the adaptation of existing activity monitors. In contrast to the typical and unassisted movements of the general population, hospital patients show different movement patterns. They typically exhibit slower movements and shorter steps and often rely on the assistance of walking aids for support, which deviate from the typical movement patterns of healthy individuals [3,20,21]. As a result, existing activity monitor algorithms are not equipped to accurately classify all aspects of physical behavior in hospital patients. Therefore, the utilization of specialized activity monitors that have been adapted and validated in the hospital population is mandatory to accurately monitor physical behavior in this population. 

Currently, only a few devices have been validated to monitor posture and motions in the hospital population, but these devices still have their limitations regarding the detection of slow walking and counting steps [22,23,24,25]. One of those devices is the Activ8 Professional Wearable Activity Sensor System (Activ8), which is a thigh-fixed wearable device that measures raw accelerations and converts them into classifications of specific postures and motions, such as time spent sitting, standing, walking, cycling, and running. Additionally, the device now provides the number of steps and the number of sit-stand transitions. The Activ8 has previously been validated in adult populations with cerebral palsy, lower limb amputations, and people after stroke with fairly good results [26,27,28]. However, also for this device, challenges were noted in accurately differentiating between standing and walking in a slow gait, and the number of steps and postural transitions were not yet available outcomes in the previous validation studies. Therefore, the existing Activ8 is less suitable for monitoring physical behavior in hospital patients. To overcome this issue, the existing Activ8 algorithms were tested and optimized for in-hospital use, resulting in the modified Activ8. As a result, the primary objective of the current study is to investigate the concurrent validity of the modified Activ8 when compared to video recordings in hospitalized patients. 

## 2. Materials and Methods

### 2.1. Optimization Process

Prior to the study of this paper, observational measurements were conducted on a convenient sample of 21 hospitalized patients during their physical therapy sessions. Because the Activ8 device does not store raw signals, the 9-degree of freedom motion logger (9DoFML, 2M Engineering, Valkenswaard, The Netherlands) was used for these measurements. This 9DoFML is a lightweight and compact wearable device that contains multiple embedded sensors (measuring 41 × 38.5 × 15.5 mm and weighing 25 g). Because the Activ8 only contains a tri-axial accelerometer, only the accelerometer signals of the 9DoFML were analyzed in this optimization process. The technical specifications of the accelerometer embedded in the 9DoFML are identical to those of Activ8. The optimization process involved an iterative process, wherein the device algorithms were tested and optimized. This refinement was based on a thorough comparison of the outcomes of video recordings and outcomes from an original Activ8 device, which was simultaneously applied. The optimization of algorithms was stopped at the moment that improvement leveled off and when it was felt that no further improvement could be reached. 

### 2.2. Study Population 

For the study of this paper, participants were recruited from the Hepato-Pancreato-Biliary Surgery or Pulmonary departments of the Erasmus MC University Medical Center Rotterdam. The inclusion criteria for the study were as follows: an age of 18 years or older, admission to hospital care units, receiving physical therapy, ability to understand and follow verbal instructions in Dutch, and a Functional Ambulation Classification (FAC) score of 3 to 5. The use of walking aids was allowed during this study. Patients with cognitive impairments or injuries to the skin or underlying tissues of the upper legs were excluded from participating. The sample size was determined based on previous validation studies conducted with the Activ8 device [26,27,28]. All participants provided informed consent by signing an informed consent form prior to participation. This study received ethical approval from the Medical Ethical Committee of Erasmus MC University Medical Center Rotterdam, The Netherlands (MEC-2020-0820).

### 2.3. Measurement Protocol 

The activity protocol encompassed a variety of basic and functional activities commonly performed by patients in a hospital setting. Each activity had a duration ranging from a minimum of 30 s to a maximum of 5 min. To ensure synchronization between the activity monitor and the video recording, the device was manually tapped at the beginning and end of each protocol activity, allowing for accurate capture of this information in the raw acceleration data. Basic activities were standardized and involved a single posture or movement, while functional activities comprised multiple postures and movements. For instance, one example of a functional activity was an undefined bathroom activity, which included a patient transitioning from the bed or chair, walking to the bathroom, standing to comb their hair or brush their teeth, and then returning to a sedentary position. Additionally, a 3 m walk test was conducted to estimate the walking speed of each participant. In cases where patients were unable or uncomfortable performing a specific activity, it was omitted from the activity protocol for this patient. Approval for participation in these activities was obtained from both their clinical physician and physical therapist after considering their circumstances.

### 2.4. Activity Monitor 

In this validation study, the modified Activ8 algorithms resulting from the optimization process were employed to transform raw accelerations into classifications of body postures, motions, position transfers, and steps in hospital patients. As in the testing and optimization measurements, we used the 9DoFML to measure the raw accelerations, because of its ability to analyze error sources and to allow better synchronization with video recordings. The raw acceleration data were collected at a sampling rate of 50 Hz. The collected data were stored within the device and subsequently downloaded using a computer connected via USB. The data was offline processed with the Activ8 algorithm to provide the algorithmic outcomes categorized, including metrics such as the time spent in sedentary postures, standing, walking, cycling, and running, along with the count of postural transfers and steps per second. To securely attach the system, it was affixed to the upper leg using medical skin tape (Figure 1), a method typically well-received by patients [27].

### 2.5. Video Recording

Video recordings were used as the criterion measure to validate body postures, motions, step count, and transfer count [29]. These video recordings were captured using the JVC Everio GZ-HM960 video camera (JVCKENWOOD Europe B.V., Mijdrecht, The Netherlands) and included the actual time in hours and seconds. The video recordings were reviewed on a monitor, where each second of the recorded activities was observed and classified into their respective classifications, such as sitting or standing, in line with predefined clinical definitions and as outlined in Table 1. Accordingly, the number of steps and position transfers during the activities were noted. This information was registered in a spreadsheet file, together with the total duration of the classification and the activity. Each recording underwent a minimum of two viewings, both at half speed, by a single researcher to ensure accuracy. A second observer analyzed the video recordings when the initial classifications were unclear. These viewings were conducted independently of the output data from the 9DoFMLo device [28]. This classification was performed independent of the protocol.

### 2.6. Data Analysis 

Data from Activ8 and video analysis were synchronized based on the researcher tapping the device during the measurement, and synchronization included a 2-s time shift of the Activ8 data to account for the Activ8 processing delay. The data analysis primarily focused on calculating the total time duration in seconds spent in various activities of the protocol, including standing, walking, being upright, and engaging in sedentary behaviors (a combined category encompassing lying and sitting), as well as quantifying the number of postural transfers and steps. These classifications were independently calculated for both the video recordings and the activity monitor output across all protocol segments, independent of the activity protocol. 

To evaluate the validity of the modified Activ8 algorithm, both absolute and relative differences were computed between the two methods. The absolute difference was determined as (Activ8—Video), while the relative difference was expressed as (absolute difference/video) × 100%. A relative difference of 10% or less at the group level was considered acceptable for validity, a criterion consistent with prior research standards [26,27,28]. If the total duration of an activity was performed or classified for less than 120 s, relative differences were not calculated. Data normality was assessed using data visualizations and the Kolmogorov–Smirnov test. To ascertain significant differences between the video recordings and Activ8 data, the Wilcoxon Signed Rank test was performed using Rstudio (version 1.4.1106, RStudio, Inc., Boston, MA, USA). A *p*-value of 0.05 was considered statistically significant. Within-subject differences are presented in Bland–Altman plots, and these plots were also assessed for potential proportional bias. 

## 3. Results

The study included 31 participants, and detailed participant characteristics are available in Table 2. Table 3 offers an overview of the frequency at which unique participants performed each activity within the protocol. The mean measurement duration that was used for analysis was 15.6 ± 3.7 min. Discrepancies in participant numbers across the different activities result from patients being unable to complete all parts of the protocol. The key findings are presented in Table 4.

### 3.1. Time Spent Walking, Standing, Upright, and Sedentary

The relative time differences between the Activ8 and the video recordings for walking, standing, and the combination of walking and standing (referred to as “upright”) fall within the acceptable limit of 10% (as indicated in Table 4). The largest relative differences between the Activ8 and the video recordings are observed in activities in the functional protocol, such as room activities (9.5%). Moreover, significant underestimations by the Activ8 were found for basic standing (−2.2%, *p* < 0.05) and upright time in the total protocol (−0.6%, *p* < 0.01). The Activ8 device significantly overestimated the time spent sitting for the whole protocol (0.4%, *p* < 0.01), the functional protocol (9.6%, *p* < 0.01), and the functional walking activity (5.8%, *p* < 0.05), with a significant underestimation for the room activity (−8.4%, *p* < 0.05). The individual differences, as illustrated in Figure 2a–d, reveal a minimal mean difference between the two methods and show no evidence of a trend indicating proportional bias.

### 3.2. Transfer and Step Count

The analysis of sit/stand transfers between the video recordings and the Activ8 device yielded non-significant relative differences of less than 7.9% for transfers and 3.2% for steps. Figure 2e,f provide insight into the individual variation between both methods and show that the mean difference or bias is slightly above zero for transfers but close to zero for steps. Importantly, there appears to be no proportional bias, with consistent differences between the methods for both figures.

## 4. Discussion

The primary objective of this study was to evaluate the concurrent validity of the modified Activ8 device for assessing postures, motions, position transfers, and step counts in hospitalized patients. This assessment was conducted in two parts: a basic protocol with prescribed postures and motions, and a functional protocol that included a wider range of functional activities. The findings from this study demonstrate that the modified Activ8 exhibits good validity in both parts of the protocol for detecting time spent walking, upright, standing, and in sedentary postures. Additionally, the new outcomes of the modified Activ8, which are the number of steps and sit-to-stand transfers, showed valid results. 

Differences between methods were defined as “acceptable” if they were within the 10-percent range at the group level. This 10-percent threshold was also used in previous research [26,27,28], while knowing that this threshold is arbitrary and that other thresholds could have been chosen. However, when using this threshold, mean differences generally fell within the defined error range and were not found statistically significant. Of the few significant differences found (such as time spent in sedentary postures), the relative differences were small and also clearly within the 10 percent acceptable error range. There were no significant differences found for the newly introduced elements of postural transitions and step count. 

An essential characteristic of this validation study is the inclusion of not only basic but also functional activities in the activity protocol. This is in agreement with the recommendations of Lindemann et al. [30]. The basic or standardized protocol allows some comparison of different monitoring methods and algorithms, but because activities are not performed in a natural way and order, they cannot be used to validate spontaneous activity in real life. Therefore, the functional protocol was added, which allows for assessing (ecological) validity in real-life conditions and increases the generalizability to real-world clinical scenarios [30]. It is noteworthy that lower validity results could be anticipated for these functional activities due to the reduced level of standardization, with more alternating and shorter durations of postures and motions, and the increased complexity of analysis involved. In line with this expectation, slightly higher relative differences at the group level were found for functional activities, although these differences remained acceptable and were mostly not statistically significant. 

In addition to examining results at the group level, the study also aimed to assess the validity of the modified Activ8 device on a per-measurement basis. These additional analyses were important, because significant random measurement errors (i.e., the occurrence of both overestimation and underestimation) can be leveled out in outcomes at the group level. The modified Activ8 indeed showed between-measurement variation that needs to be considered. First of all, the results showed that the variability depends on the type of outcome: the analyses revealed larger between-measurement variation for metrics such as time spent standing, time spent walking, and the number of steps, and smaller variation for upright and sedentary time. Secondly, as the results generally do not indicate proportional bias, measurement errors are relatively more pronounced at activities with low numbers or short durations. This indicates that, especially in cases where time spent in activities or the number is low, the absolute error is still small, but the relative difference might be considerable. For example, as illustrated in Figure 2a, the Bland–Altman plot reveals a more pronounced variation between measurements for shorter walking periods, characterized as ‘room activities’, in contrast to the relatively smaller variation observed during the majority of longer walking periods. Many factors might influence the between-measurement variability, and one of the potential factors for errors in time spent walking and number of steps might be the walking speed and the use of walking aids. It can be expected that the validity of measurement is lower in people with lower walking speeds and/or using aids [23,31]. In post hoc analyses, the potential impact of individual variations in walking speed and use of walking aids on the data were explored, but no clear effect was found. Nevertheless, the limited accuracy of the modified Activ8 needs to be considered when applied to individual patients and when outcomes have low values. 

Despite the standardized part of the protocol, determining the validity of the modified Activ8 compared to other potentially applicable devices in the hospital setting is a challenge. Direct comparison is difficult due to differences in activity protocols, outcome measures, and analyses [30]. One of the solutions for this is simultaneously examining multiple devices, which allows direct comparison. For practical reasons, this was not possible in the current study, but the simultaneous validation of multiple devices, including the modified Activ8, would be a meaningful next step to enable device comparison. Currently, several other single-sensor systems have been validated for identifying specific postures and motions, transfers, or steps in hospitalized patients, including Activpal, Actigraph, MOX, Dynaport, and SENS [23,25,31,32,33,34]. However, it was found that most of these devices still face challenges in accurately detecting specific postures or motions, such as walking [23], standing [23,34], upright activity, or sedentary activity [34]. Furthermore, previous research also described problems with accurate step count assessment due to variations in walking speed and step length, especially for patients with a lower walking speed [23,31]. From that perspective, the results of the study support the potential advantages of the modified Activ8 [23,33]. 

A somewhat easier comparison can be made when referring to previous Activ8 validation studies because of similarities in protocol, measures, and analyses [26,27,28,35]. The optimization of the previous Activ8 algorithm was partially motivated by its lower performance in the detection of walking, particularly in cases of slow walking [27], which is not an uncommon issue for accelerometers with thigh placement [36]. Particularly promising is that the results of the modified Activ8, both at the group and individual levels, appear to be superior to those of the non-optimized Activ8 algorithm, with detection of slow walking no longer being an issue [27,28]. This suggests that the modified Activ8 demonstrates promising validity results for monitoring activities and postures in clinical settings. 

This study has several potential limitations that should be acknowledged. Firstly, in this study, the 9DoFML and not the Activ8 itself was used to collect high-resolution raw acceleration data. To ensure comparability between the output from the modified Activ8 and the 9DoFML, multiple simultaneous measurements were conducted with both devices, and they all found similar outcomes. Secondly, using video recordings as a reference method is both a strength and a limitation [29]. Activity definitions and observer judgments can introduce variations in the scoring of video recordings. For example, distinguishing between walking, shuffling, and ‘standing with movement’ can be challenging, depending on an individual’s walking ability. To ensure accuracy, a second observer analyzed the video recordings when the initial classifications were unclear. A notable strength of this study, however, is the inclusion of a wide variety of hospital patients from medical and surgical wards, encompassing variations in walking speed and the use of various walking aids. As a result, the included participants are assumed to represent various hospital subpopulations, enhancing the generalizability of the findings to a broader clinical context. 

Given the challenges associated with comparing the modified Activ8 to other potentially applicable devices within a hospital setting, several options for future research and development have become evident. Firstly, there is a critical need for standardization in activity protocols, outcome measures, and analytical approaches to facilitate more direct and meaningful comparisons across devices. As mentioned earlier in this discussion, comparison among devices remains difficult and should be facilitated by validation studies including multiple devices. This would help to comprehensively understand a device’s strengths and limitations, enabling researchers to identify the most suitable device for their targeted patient population. Moreover, given the complexity of accurately detecting specific postures, motions, postural transitions, and steps, especially in people with distorted movement patterns, future research should continue to prioritize the refinement of algorithms and methodologies. It encompasses addressing current issues like assessing outcomes such as walking speed or step length, but should also focus on differentiating between the various sedentary postures, which is relevant when monitoring physical behavior in hospital patients. This effort of ongoing algorithm refinement aligns with the continuous technological advancements in the field and should consider following the existing FAIR principles to benefit data sharing and open science [37]. Additionally, considering the rapidly evolving landscape of wearable sensor technologies, ongoing efforts should be directed towards the continuous improvement, innovation, and finally implementation of devices like the modified Activ8, ensuring their relevance and efficacy in monitoring and assessing patient activities in diverse healthcare settings. Overall, these future recommendations aim to advance the field, enhancing the accuracy, reliability, and applicability of wearable sensors in healthcare contexts.

## 5. Conclusions

In conclusion, the modified Activ8 is a valuable device for assessing the physical behaviors of hospitalized patients. The device covers a large set of relevant outcomes with good validity results. Only for less frequent activities and postures and motions of short duration, both over the whole assessment period, do outcomes have to be interpreted with caution. 

## Figures and Tables

**Figure 1 sensors-24-00180-f001:**
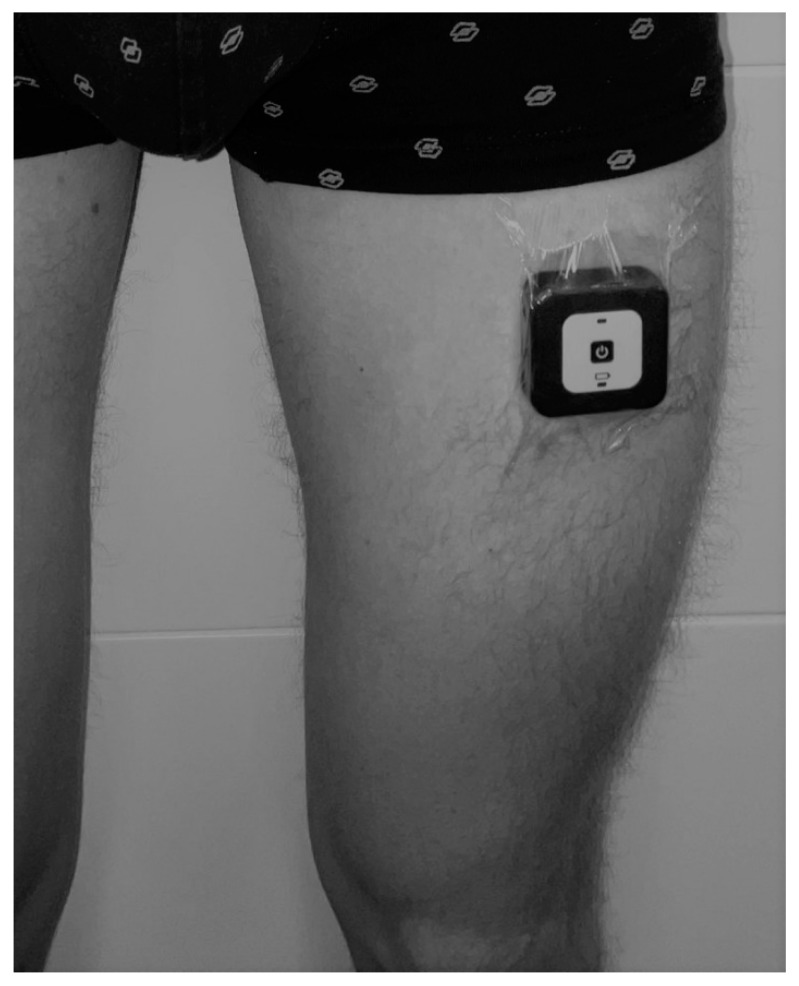
Positioning of 9DoFML device on upper thigh using medical skin tape.

**Figure 2 sensors-24-00180-f002:**
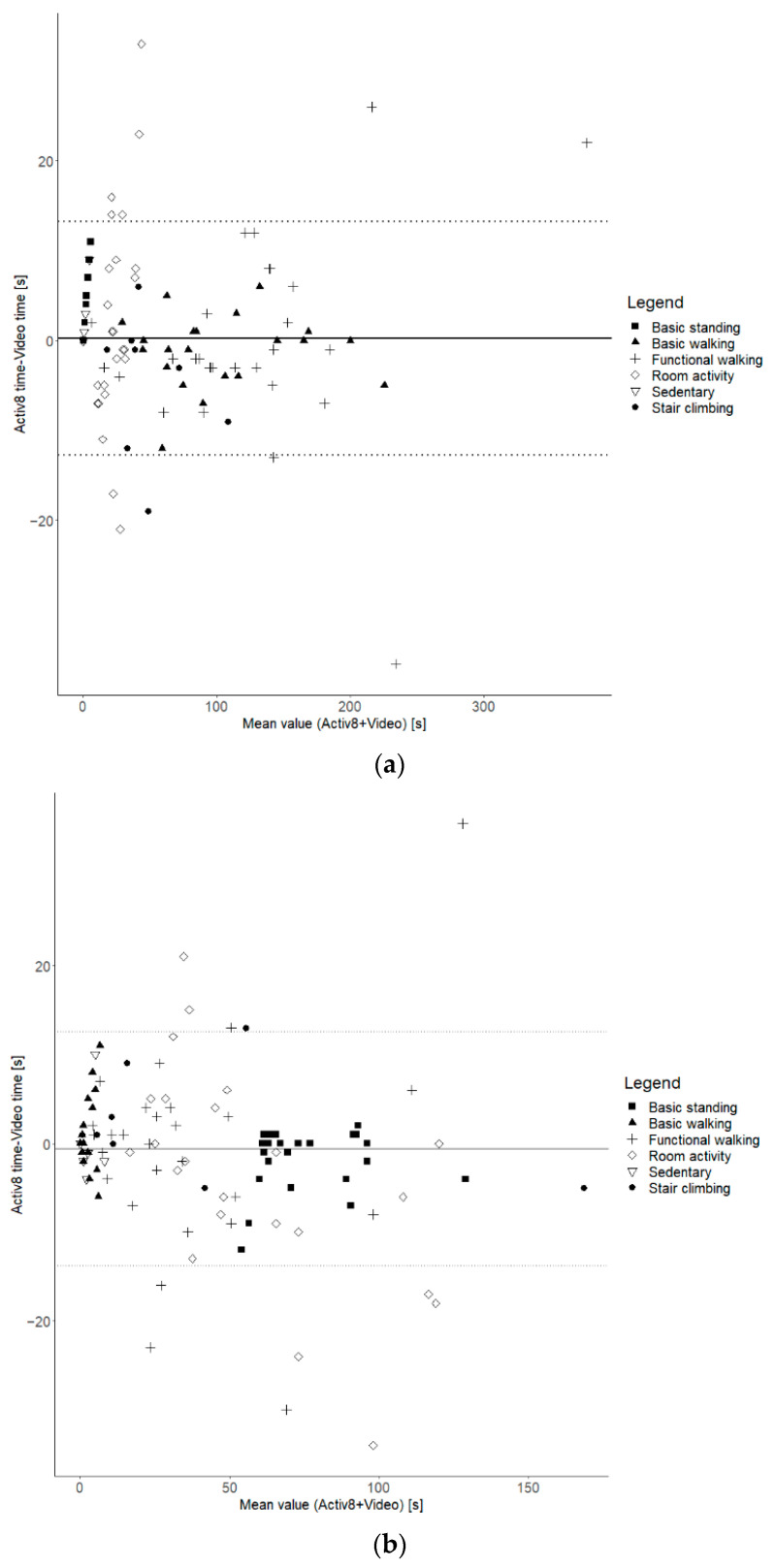
Bland–Altman plots. (**a**) Time spent walking for basic + functional protocol; (**b**) time spent standing for basic + functional protocol; (**c**) time spent upright for basic + functional protocol; (**d**) time spent sedentary for basic + functional protocol; (**e**) transfers for functional protocol; (**f**) steps for basic + functional protocol.

**Table 1 sensors-24-00180-t001:** Translation between video recordings and Activ8 classifications for postures and motions.

**Video**	Lying	Sitting	Sitting with movement	Sit-to-stand transfer	Standing	Standing with movement	Shuffling	Walking	Stair climbing	Cycling
**Activ8**	Sedentary			Standing			Walking			Cycling

**Table 2 sensors-24-00180-t002:** Patient characteristics.

Sample Size	31
Age (m, sd)	54.0, 17.0
Gender (m/f)	21/10
Medical Department HPB/Lung	18/13
Surgical/Medical	18/13
Walking aid (Walker/IV pole/none)	10/12/9
Days since admission (m, sd)	23.0, 37.3
Walking speed on 3 MWT (m, sd) [km/h]	2.10, 0.82

**Table 3 sensors-24-00180-t003:** Number of observations for each part of the protocol.

Protocol	Activity	*n*
Basic	Walking	23
	Standing	27
	Sitting on a chair	30
	Sitting on the edge of the bed	25
	Lying	28
Functional	Hallway activities	26
	(Bath)room activities	24
	Stair climbing	8

**Table 4 sensors-24-00180-t004:** Summed data of all subjects and between-subject ranges for both protocols separately and joined.

	Type of Protocol	Type of Activity	Total Video	Total Activ8	Absolute Difference	Relative Difference (%)
Walking time ^1^ [s]	Total		6694	6741	47	0.7
	Basic	Walking	2205	2181	−24	−1.1
	Functional	Total	4489	4497	8	0.2
		Walking	3518	3512	−6	−0.2
		Room	557	610	53	9.5
		Stair climbing	414	375	−39	−9.4
Standing time ^1^ [s]	Total		4806	4693	−113	−2.4
	Basic	Standing	2050	2005	−45	−2.2 *
	Functional	Total	2695	2608	−87	−3.2
		Walking	1004	979	−25	−2.5
		Room	1391	1323	−68	−4.9
		Stair climbing	300	317	17	5.7
Upright time ^1^ [s]	Total		11,500	11,434	−66	−0.6 **
	Basic	Walking + standing	4289	4295	6	0.1
	Functional	Total	7184	7105	−79	−1.1
		Walking	4522	4491	−31	−0.7 #
		Room	1948	1922	−22	−3.1
		Stair climbing	714	692	−22	−3.1
Sedentary time ^1^ [s]	Total		17,478	17,544	66	0.4 **
	Basic	Total	16,428	16,391	−37	−0.2
		Sitting on chair	5027	5029	2	0.0
		Sitting on bed	2865	2870	5	0.2
		Lying	8492	8536	44	0.5
	Functional	Total	1047	1147	100	9.6 **
		Walking	530	561	31	5.8 **
		Room	610	559	−51	−8.4 *
		Stair climbing	4	27	23	^NA^
Transfers [nr]	Functional	Total	98	101	3	3.1
		Walking	51	55	4	7.8 #
		Room	46	45	−1	−2.2
		Stair climbing	1	1	0	0.0
Steps [nr]	Total		8228	8214	−14	−0.2
	Basic	Walking	2823	2782	−41	−1.5
	Functional	Total	5403	5360	−43	−0.8
		Walking	4144	4237	93	2.2
		Room	724	710	−14	−1.9
		Stair climbing	442	456	14	3.2

^1^ Time represents the cumulative time of all participants in seconds. * *p*-value < 0.05; ** *p*-value < 0.01; # *p*-value < 0.1. ^NA^ Not applicable because of the short duration.

## Data Availability

The data are not publicly available due to privacy/ethical restrictions.

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
