# Peer review of "Validation of the Activ8 Activity Monitor for Monitoring Postures, Motions, Transfers, and Steps of Hospitalized Patients"

_sensors, 2023, doi:10.3390/s24010180_

Round 1

Reviewer 1 Report

Comments and Suggestions for Authors

Thank you for the opportunity to review this very well written manuscript outlining the evaluation of the modified Activ8 for inpatient use.

In my opinion the research is sound, but the structure of the manuscript could be improved and some more detail needed for some methods.

Major Changes:

More detail is needed in the Data Analysis section. How is duration being calculated and by what unit? (seconds/milliseconds/other). More detail needed on the video scoring system or methods used for scoring duration by the observer. Unclear if the observer was given a copy of the protocol and asked to time the different activities, or identify activities and durations independent of protocol, if this was done on paper, digitally etc.

Materials and Methods: The order of the subsections within need to be rearranged to aid in understanding. For example, the Optimisation Process is outlined referencing the device and the video recordings. However these subsections with required details do not appear until later in the methods. It is also not very clear if the Optimisation Process took place before this study, or as part of this study.

Line 129: the 9DOFML device that was used in place of the modified Activ8 - you mention that it is from the same manufacturer, but is it the same technology used within the Activ8? if it is not the same technology more detail/justification is needed to draw parallels.

Minor Changes:

Line 17: should this be modified Activ8 and not revised Activ8? Same with line 25 "improved Activ8"

Line 37 "interventions such as mobilization"

Line 88 is the first time the full name of the modified Activ8 is introduced, but the shortened title (modified Activ8) had been used up to this point. Needs to be introduced with full name in first instance.

Paragraph starting at line 126 uses first person voice - all others use third. Should change to third person for consistency.

Line 20 & 22 in abstract also in first person voice.

Line 210: remove line break/new line

Line 289: should this be "the modified Activ8"

Author Response

We are sincerely grateful for your thoughtful and constructive feedback, which has significantly contributed to the refinement of our manuscript. Your careful reviews have been instrumental in enhancing the overall quality of our work. We are committed to incorporating each of the valuable suggestions into the revised manuscript.

We believe that the proposed changes will not only address the specific concerns about the methods and structure that were raised, but will also substantially strengthen the overall contribution of our research to Sensors. We appreciate the time and expertise you have dedicated to the review process and look forward to the opportunity to share the improved version of our study with the journal's readership.

Thank you once again for your insightful comments and constructive feedback. In our response to your reviewer comments we refer to the line numbers as mentioned in the version of the manuscript including tracked changes. Additionally, we provide a manuscript without tracked changes. If there are any questions or additional comments please let us know.

Warm regards,

Marlissa Becker and co-authors.

Reviewer 2 Report

Comments and Suggestions for Authors

Author Response

We are sincerely grateful for your thoughtful and constructive feedback, which has significantly contributed to the refinement of our manuscript. Your careful reviews have been instrumental in enhancing the overall quality of our work. We are committed to incorporating each of the valuable suggestions into the revised manuscript.

We believe that the proposed changes will not only address the specific concerns about the methods and structure that were raised, but will also substantially strengthen the overall contribution of our research to Sensors. We appreciate the time and expertise you have dedicated to the review process and look forward to the opportunity to share the improved version of our study with the journal's readership.

Thank you once again for your insightful comments and constructive feedback. In our response to your reviewer comments in the attachment, we refer to the line numbers as mentioned in the version of the manuscript including tracked changes. Additionally, we provide a manuscript without tracked changes. If there are any questions or additional comments please let us know. 

Warm regards,

Marlissa Becker and co-authors.
